

# Robust poleward jet shifts in idealised baroclinic-wave life-cycle experiments with noisy initial conditions

Felix Jäger[1,2], Philip Rupp[1], and Thomas Birner[1]

[1]Meteorological Institute Munich, Ludwig-Maximilians-University, Munich, Germany
[2]Now at Institute for Atmospheric and Climate Science, ETH Zurich, Zurich, Switzerland
**Correspondence:** Philip Rupp (philip.rupp@lmu.de)

**Abstract.**

Idealised baroclinic-wave life-cycle experiments are a widely used tool to study fundamental characteristics of mid-latitude baroclinic instability. A typical life-cycle evolves from an initialised baroclinically unstable jet through an exponential growth phase of a particular unstable wave mode, followed by wave breaking during the mature phase, and wave-mean flow interaction driving a jet shift during the decay phase. Many authors distinguish between life-cycles with predominantly anticyclonic (LC1) and cyclonic (LC2) wave breaking and the transition between the two flavours is typically controlled via the strength of cyclonic meridional wind shear in the initial conditions. While baroclinic wave growth has traditionally been triggered via a specified initial perturbation with fixed zonal wave number, this study extends the concept of baroclinic-wave life-cycles by analysing the influence of random initial perturbations without any preferred zonal dependency on the life-cycle evolution. We find that the growth phase shows a robust LC1-LC2 distinction as a function of initialised meridional shear, while a preference for LC1-like characteristics is observed during the decay phase for all life-cycles with non-monochromatic initial perturbations. In particular, the persistent cut-off cyclones that typically form for LC2 initialisations are found to eventually become unstable - the earlier during the life-cycle the stronger the initial noise perturbations. All non-monochromatic life-cycles result in a poleward jet shift in their final state, regardless of the strength of the initial shear. Consistently, anticyclonic wave breaking tends to be predominant during the mature and decay phases, even for LC2 initialisations. Equatorward jet shifts associated with cyclonic wave breaking still exist, although purely as a transient interim state. We show that wave-wave interactions resulting from the initialised random wave spectrum play an important role during all phases of the life-cycle.

## 1 Introduction

The large-scale cyclones and anticyclones that dominate mid-latitude weather ultimately arise as a result of baroclinic instability driven by the equator-to-pole temperature contrast on a fast rotating planet. Early theoretical insights of the fundamental process of baroclinic instability were based on linear theory (Charney, 1947; Eady, 1949). A more complete understanding was achieved by non-linear model experiments that revealed the typical life-cycle of a baroclinically unstable wave: a quasi-linear growth phase is followed by wave breaking and non-linear decay (Simmons and Hoskins, 1978). Notably these non-linear experiments demonstrate the important role of baroclinic wave breaking in the exchange of energy and momentum between



lower and higher latitudes and show how wave-mean flow interactions during the non-linear decay phase lead to meridional shifts of the initial jet. Furthermore, the relevance of isolated life-cycles with pronounced wave growth and decay phases for the atmosphere has been established observationally (e.g., Randel and Stanford, 1985a, b).

Past research has identified two distinct flavours or paradigms of idealised baroclinic-wave life-cycle evolution, associated with predominantly anticyclonic (LC1) or cyclonic (LC2) wave breaking (Simmons and Hoskins, 1980; Thorncroft et al.,
1993). Which life-cycle flavour is observed during an experiment depends on the details of the initial conditions and is typically controlled via the strength and direction of the meridional wind shear in the specified initial jet. The evolution of these two life-cycle flavours distinctly differs in terms of their energy conversions and potential vorticity (PV) signatures. In addition, LC1 is associated with mostly poleward eddy momentum fluxes and a corresponding poleward jet shift, while LC2 exhibits mostly equatorward momentum fluxes and hence an equatorward jet shift. Observational studies found a systematic poleward
propagation of zonal mean flow anomalies associated with wave breaking events (Feldstein, 1998), suggesting a tendency of the real atmosphere to favour anticyclonic breaking (also see Lee et al., 2007).

Several authors have shown that the flavour of life-cycles in idealised experiments can strongly depend on the initial zonal wave number $k$ of the perturbation used to trigger the growth of the baroclinic wave (Hartmann and Zuercher, 1998; Orlanski, 2003; Wittman et al., 2007). While higher wave numbers favour LC2 evolution irrespective of the initial basic state, lower wave
numbers tend to produce LC1 behaviour. For certain intermediate zonal wave numbers (usually $k \approx 6$) the life-cycle flavour strongly depends on the details of the baroclinically unstable initial jet and the specifics of the system. In that sense a transition between predominantly LC1 and LC2 characteristics can be forced via, e.g., the inclusion of moist processes (Orlanski, 2003), varying meridional wind shear (Thorncroft et al., 1993) or altered stratospheric conditions (Wittman et al., 2007; Kunz et al., 2009). Note that a striking commonality of most previous life-cycle studies is the use of a monochromatic initial perturbation
with a single specified zonal wave number, while the evolution of the real atmosphere generally involves a range of wave numbers.

The present manuscript aims to extend the existing literature by investigating the evolution of baroclinic-wave life-cycles for non-monochromatic initial perturbations and the corresponding robustness of typical LC1 and LC2 characteristics. In section 2, details on the numerical model and the initial conditions are given and the general approach of this study is described. Section 3
then discusses numerical simulations where the initial perturbation is dominated by a single zonal wave number and contains no or only weak additional noise. Simulations with noise-dominated initial perturbations are described in Section 4. Some main findings of this study are discussed in section 5 and a summary of our key conclusions is given in section 6.

## 2 Details on the numerical model and initial conditions

All experiments in this study are performed with the simple dry dynamical core model BOB (Built on Beowulf; see Rivier et al.,
2002). The model integrates a spectral version of the primitive equations on a sphere with triangular truncation at horizontal wave number 85. The discrete vertical pressure levels are distributed with constant spacing of $\Delta z = 250$ m up to a height of $z = 60$ km, where $z = -H\ln(p/p_0)$ is a log-pressure coordinate with scale height $H = 7.5$ km and reference pressure





$p_0 = 1000$ hPa. We further added a sponge layer consisting of 10 additional model levels between $z = 60$ km and $z = 82$ km, equally spaced in pressure. The bottom boundary is given by the pressure surface $p = 1000hPa$ without any orographic

forcing. The model time step is 5 minutes and all experiments are integrated for 45 days of model time to fully cover the evolution during the late stages of all life-cycles and to ensure the model reaches a steady final state. Most analyses shown in this manuscript are based on daily output of instantaneous fields, however, some figures will show 6-hourly output for a more detailed representation of the dynamical evolution. To ensure numerical stability and energy dissipation via sub-grid-scale processes, the model includes a sixth order hyper-diffusion, damping the largest resolved wave numbers on a timescale of $2.4$

65   h.

Experiments are initialised with a zonally symmetric state describing a baroclinically unstable jet and a superimposed small perturbation to trigger the instability (see Appendix for technical details). The zonally symmetric initial state corresponds to either LC1 or LC2 initial conditions, shown in Fig. 1. Note that these initial states are similar[1] to the ones used in Thorncroft et al. (1993) and hence favour predominantly anticyclonic and cyclonic wave breaking, respectively, for initial perturbations

with a single zonal wave number $k = 6$. In this study we show that the model behaviour is substantially modified if the initial perturbation is non-monochromatic (i.e., consists of more than one zonal wave number) and the resulting life-cycles often deviate strongly from the canonical paradigms. We will therefore use the nomenclature "LC1" and "LC2" to refer only to the initial state configuration of the corresponding experiment. We further performed sensitivity experiments with initial states characterised by varying values of the meridional shear parameter $\hat{U}_s$ and found our results to be overall robust.

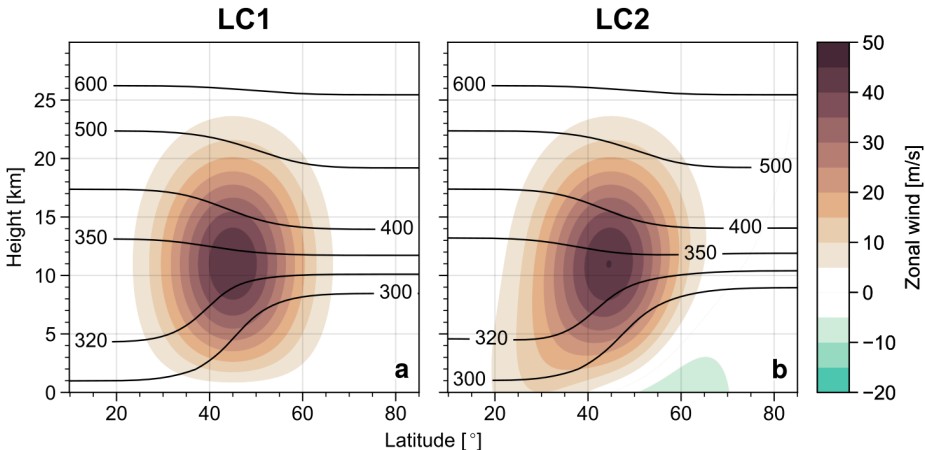

**Figure 1.** Zonal mean zonal wind (shading) and potential temperature (contours, in K) for (a) LC1 and (b) LC2 initial conditions.

The temperature perturbation (following, e.g., Rupp and Birner, 2021) used to trigger the baroclinic instability consists of two components: a monochromatic wave component with a single zonal wave number $k = 6$ and magnitude $T_w$, and a random noise component of magnitude $T_n$ (see Appendix for details). Fixed zonal wave number perturbations with $k = 6$ are a typical

---

[1]Thorncroft et al. (1993) do not give details on how their initial conditions were constructed.





choice within life-cycle studies (Thorncroft et al., 1993; Magnusdottir and Haynes, 1996; Polvani and Esler, 2007; Kunz et al., 2009; Rupp and Birner, 2021). We further introduce a noise strength as the ratio $\eta = T_n/T_w$, which is the main parameter varied in the present study to systematically transition from monochromatic initial perturbations with specified zonal wave number ($\eta \ll 1$) to noisy initial perturbations without any specific zonal dependence ($\eta \gg 1$). Experiments with $\eta > 0$ are performed as ensembles of several different noise-realisations, although the overall large-scale evolution seems insensitive to the particular details of the realisation.

## 3 Life-cycles with weak initial noise strength ($\eta \ll 1$)

This study aims to investigate how the evolution of baroclinic-wave life-cycles is modified when the initial conditions are perturbed with random noise, in contrast to the more traditional perturbation with a single specified zonal wave number. To do so, we follow a perturbation approach with gradually increasing noise strength $\eta$ from single-wave dominated ($\eta \ll 1$) to noise dominated ($\eta \gg 1$) systems and analyse the corresponding changes in certain life-cycle characteristics.

Figure 2 compares the energetics of life-cycles with LC1 and LC2 initial conditions and initial perturbations with either a single wave number ($\eta = 0$) or a specified wave number and weak superimposed initial noise ($\eta = 10^{-3}$). During the first 20 days experiments with and without noise show almost identical evolution in terms of eddy kinetic energy (EKE) and mean kinetic energy (MKE) and generally follow the respective canonical characteristics of LC1 and LC2 simulations (Simmons and Hoskins, 1980; Thorncroft et al., 1993). In particular Fig. 2 displays pronounced phases of wave growth (associated with an increase in EKE) and decay (associated with decrease in EKE) and a corresponding conversion of EKE to MKE during the later stages of the life-cycle. A common characteristic of the canonical LC2 evolution without initial noise is the elevated level of final state EKE and the relatively weak increase in MKE during the life-cycle (compared to LC1). A set of sensitivity experiments with perturbations of varying wave numbers and without noise (not shown) suggests that the difference in initial growth rates between LC1 and LC2 experiments is mostly a result of the modified initial state. The growth rate changes rather gradual with varying initial meridional shear ($\hat{U}_s$) and should not be regarded as a distinctive characteristic of the life-cycle flavour itself. The level of final state EKE, on the other hand, shows an abrupt change at $\hat{U}_s \approx 7$ m/s in these monochromatically perturbed experiments and is therefore typically a useful diagnostic to differentiate between life-cycle behaviours.

After about 20 days the experiments with weak initial noise start to diverge from the corresponding monochromatic runs. Both LC1 and LC2 enter a second phase of wave growth and subsequent decay, hence showing a peak in EKE at about day 22. However, while this noise-induced EKE peak is relatively weak for LC1 it is very pronounced and even exceeds the primary peak (day 12) in the LC2 case. Following the noise-induced wave breaking LC2 EKE drops to values lower than in the original LC2 final state and comparable to LC1 final conditions. The decrease in EKE is associated with a barotropic conversion of EKE to MKE at about day 25 and a corresponding increase in MKE[2].

---

[2]The noise-induced wave breaking is also associated with a corresponding decrease of mean potential energy (MPE) and a pronounced peak in eddy potential energy (EPE), closing the typical flow of energy from MPE to MKE via EPE and EKE (not shown).



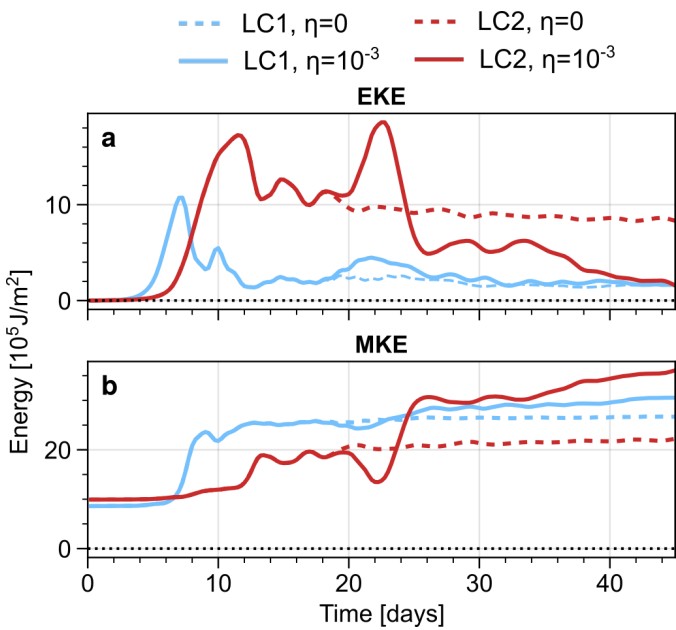

**Figure 2.** Evolution of (a) eddy kinetic energy and (b) mean kinetic energy for LC1 and LC2 initialisations, each without ($\eta = 0$, dashed) and with weak ($\eta = 10^{-3}$, solid) initial noise. Energies are shown as northern-hemispheric horizontally averaged and vertically integrated energy densities.

The introduction of weak initial noise modifies the dynamics of the system in the late stages of the life-cycle, which is also reflected in corresponding changes of the PV distribution (Fig. 3). Following their typical evolution, LC1 and LC2 experiments

exhibit phases of pronounced anticyclonic and cyclonic wave breaking, respectively, around the time of the primary maximum in EKE (cf. Fig. 2, ca. day 8 for LC1 and day 11 for LC2). The breaking and decay phases of the life-cycles are then followed by periods of quasi-steady "final states", characterised by weak zonal perturbations for LC1 and pronounced isolated cyclonic vortices for LC2 (Figs. 3b and f). Due to the dominant wave number of the imposed initial perturbation both experiments show a clear zonal periodicity with wave number 6 during this primary cycle.

Consistent with the energetics of the systems, an additional noise-induced cycle of pronounced wave growth and breaking occurs at about day 22 (Figs. 3c and g). However, this time the PV structures of both LC1 and LC2 experiments display signs of anticyclonic wave breaking with dominant zonal wave number 4. The corresponding waves decay rapidly in both cases and a steady and almost zonally symmetric final state is reached (Figs. 3d and h). It should be noted that despite the lack of obvious coherent vortices in the LC2 at case day 34 the actual final states of both life-cycles still differ in terms of the details in their

PV distribution and the corresponding meridional PV gradient. While LC1 develops a pronounced surf zone south of the final state jet (associated with sharp PV gradients near 40° and 60° latitude), LC2 is characterised by reduced PV values north of the final jet. This behaviour is consistent with the evolution and direction of wave breaking during the primary life-cycle and can be associated with stirring and mixing on the respective side of the jet (equatorward for LC1 and poleward for LC2).



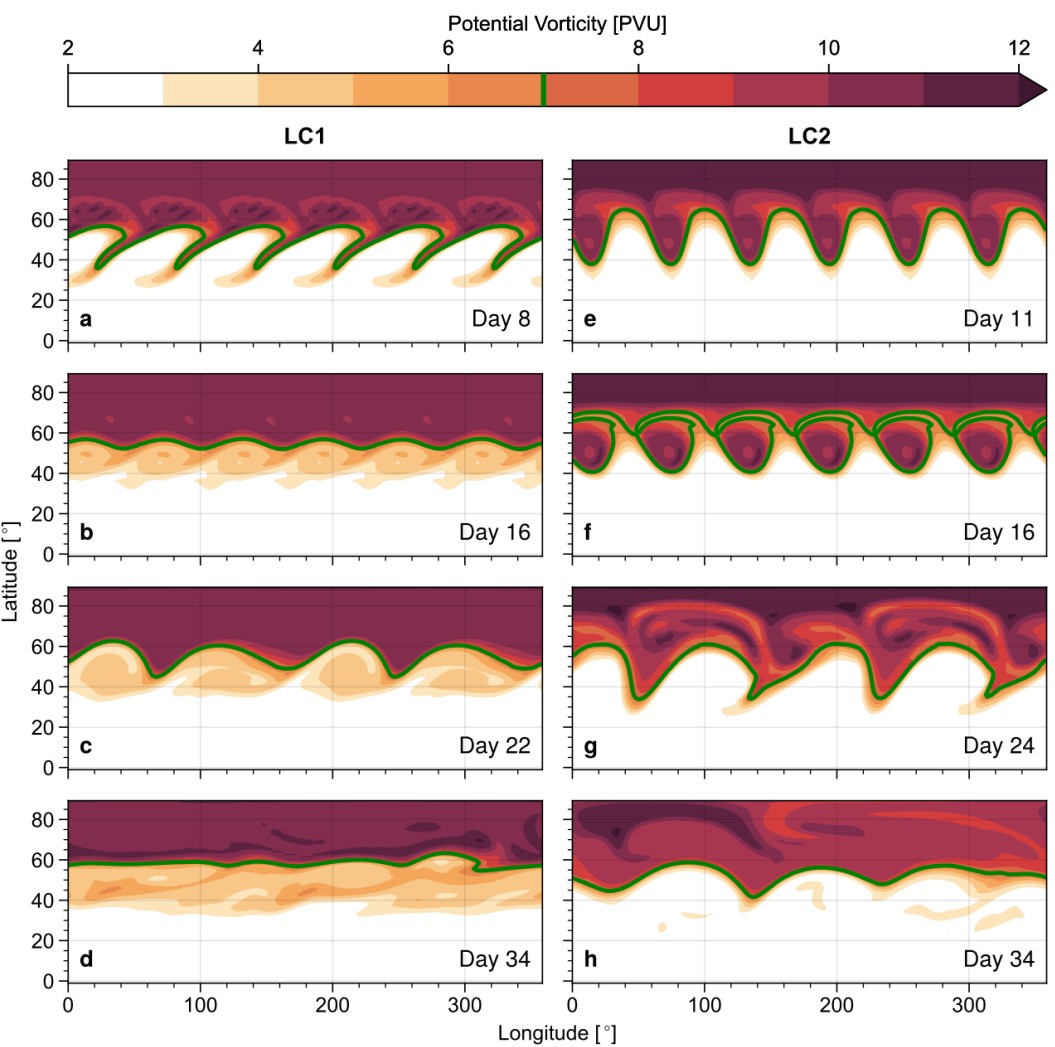

**Figure 3.** Potential vorticity on the 350 K isentrope at different days for LC1 (a-d) and LC2 (e-h) initial conditions and weak initial noise perturbations ($\eta = 10^{-3}$). The 7 PVU contour is highlighted as visual aid.





Changes in the PV distribution are also associated with changes in the wind field; in particular, a shift in the meridional position of the jet over the course of the life-cycle. The 10 km zonal mean zonal wind profiles of the final state for experiments with no initial noise (Fig. 4) show the typical poleward shift of the jet core for LC1 initial conditions compared to the initial state, while LC2 conditions lead to a slight equatorward shift throughout the life-cycle. The cyclonic wave breaking and associated equatorward eddy momentum fluxes (shown in Fig. 5) during the LC2 life-cycle further lead to negative zonal winds at about 60°.

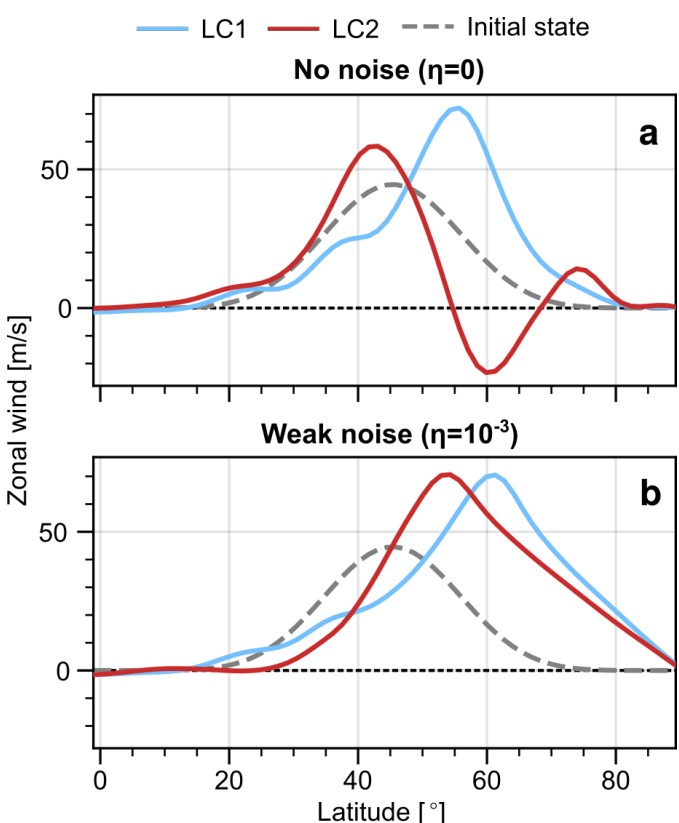

**Figure 4.** Zonal mean zonal wind at 10 km averaged over the final state (days 40-45) for LC1 and LC2 initial conditions without noise (a, $\eta = 0$) and with weak noise (b, $\eta = 10^{-3}$). Dashed profiles show the corresponding winds of the initial conditions.

In the case with weak initial noise (Fig. 4b) the final state jet profiles for LC1 and LC2 initialisations are much less distinct and both show a pronounced poleward shift. In the LC1 situation, the noise-induced wave breaking (cf. Fig. 3) leads to a slight additional poleward shift of the jet core and a substantial strengthening of the winds north of about 60°. Although the LC2 run exhibits a slightly weaker net-poleward jet shift throughout the life-cycle compared to the LC1 counterpart, the 10 km zonal wind of the final state is eastward at all latitudes. These changes in the wind profile are consistent with the robustly anticyclonic wave breaking observed during the noise-induced peak in EKE and corresponding increases in MKE (Figs. 2 and 3).



The introduction of weak noise in the initial conditions leads to substantially less pronounced differences between the final states of LC1 and LC2 initialisations compared to the clear dichotomy observed for monochromatic perturbations with a single zonal wave number. As elaborated above, this change is mainly due the occurrence of a noise-induced phase of anticyclonic wave breaking following the primary cycle. Recall that the primary cycles of LC1 and LC2 are typically characterised by dis-

140 tinct anticyclonic and cyclonic wave breaking, respectively, and the direction of wave breaking can be linked to the direction of meridional eddy momentum fluxes (Thorncroft et al., 1993) and corresponding shifts in the jet. Hence systems with LC2 initial conditions and weak initial noise experience a transient phase of equatorward jet shift. Since the noise-induced anticyclonic cycle for LC2 initial conditions occurs earlier for stronger initial noise perturbations (corresponding to larger values of $\eta$ if $T_w$ is fixed), this leads to a direct correlation between the life time of phases with equatorward shifted jet (due to the cyclonic

primary wave breaking) and the strength of the initial noise.

Figure 5a displays the evolution of the hemispheric mean meridional eddy momentum flux $[u'v']$ for LC2 experiments with varying noise strength $\eta$. All cases with weak noise ($\eta < 1$) show negative momentum flux between days 10 and 14, consistent with the period of primary cyclonic wave breaking seen in Figs. 2 and 3. A following period with vanishing momentum flux (associated with the preliminary LC2 final state) leads to a quasi-steady phase with negative accumulated momentum

flux (Fig. 5b) and a corresponding equatorward jet shift. However, as mentioned before, this phase is of transient nature and becomes systematically shorter for stronger initial noise. Figure 5c shows directly how the time of emergence of a poleward jet shift (corresponding to positive accumulated momentum flux) converges towards the time of maximum equatorward shift of the jet as $\eta \to 1$ and hence initial perturbations become noise-dominated. As will be discussed later, initialisations without any preferred zonal wave number show a robust evolution characterised by consecutive phases of equatorward and poleward

momentum fluxes independent of the initial state. On the other hand, Fig. 5c suggests that the period with equatorward shifted jet persists arbitrarily long for small enough $\eta$ and the jet profile shown in Fig. 4a therefore indeed represents a steady, yet highly unstable, final state of LC2 initialisations.

Next we investigate how the breakdown of this metastable state in LC2 simulations takes place dynamically. In particular we want to understand to what extent the increased potential for non-linear wave-wave interactions due to the introduction of

160 initial noise might explain certain details of the dynamical evolution discussed in this section. Figure 6 shows the evolution of EKE for individual wave components with varying zonal wave number $k$ and $\eta = 10^{-3}$. Both LC1 and LC2 experiments with weak initial noise show different stages characterised by different types of wave interaction. During the very early phase the system is dominated by the linear growth of wave 6 associated with the primary life-cycle. Waves with $k \neq 6$, however, grow rapidly with growth rates almost independent of their wave number. Note that the associated growth rates of these waves seem

to be significantly larger than the growth rates in reference simulations with initial perturbations of only the respective single wave number and no noise (cf. Fig. S1 in the Supplement). This accelerated growth suggests a non-linear interaction of the different wave components and appears to be similar to quasi-linear non-normal growth (e.g., Farrell and Ioannou, 1996).

With the onset of the primary wave breaking (about day 8 for LC1 and day 12 for LC2 initialisations) different wave components start to behave differently. While waves with wave numbers 5, 3 and 1 experience a substantial reduction in

growth rate, wave numbers 4 and 2 continue to grow at elevated rates. This filtering of zonal wave components is consistent

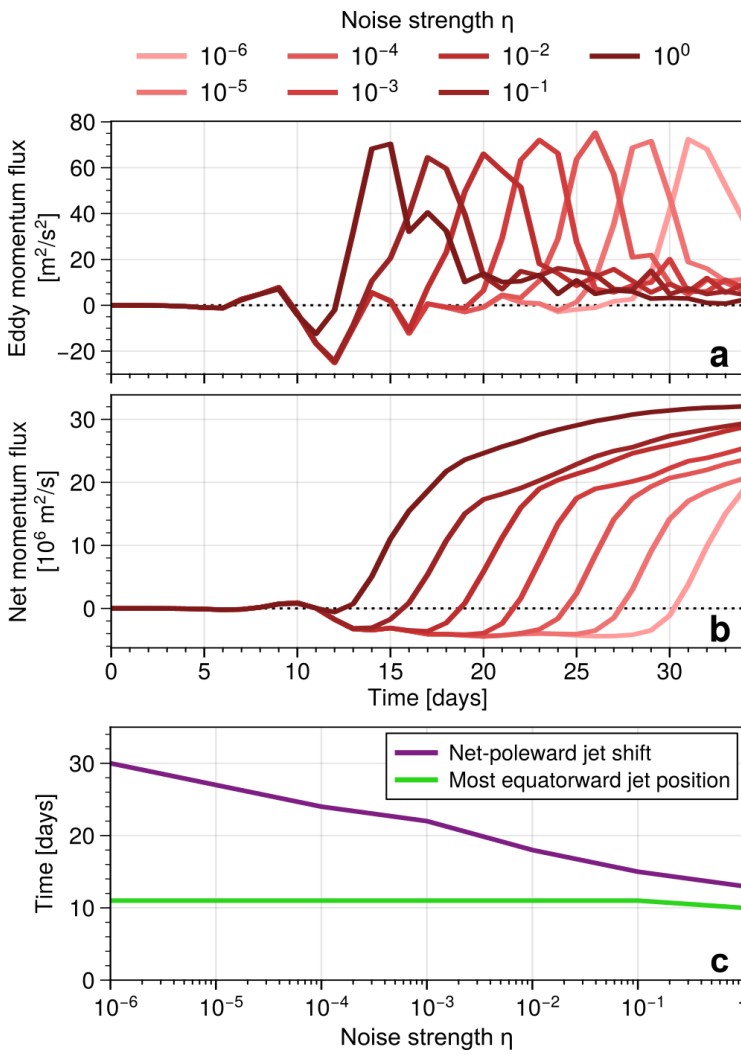

**Figure 5.** Evolution of (a) hemispherically averaged eddy momentum flux and (b) accumulated net eddy momentum flux for LC2 initial conditions and different values of noise strength $\eta$. Panel c: time of most equatorward shifted jet (maximum of the zonal mean zonal wind profile at 10 km), as well as the time when the corresponding wind maximum is located north of the initial wind maximum ($45°$) for the first time during the experiment. All lines show the mean over three realisations of initial noise.

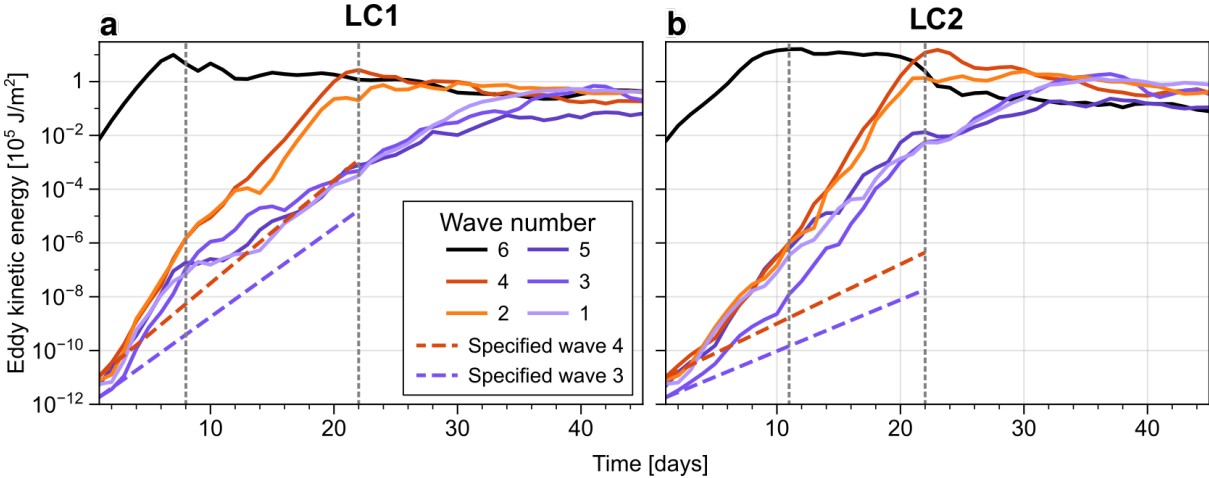

**Figure 6.** Evolution of EKE for selected wave numbers in experiments with LC1 and LC2 initial conditions and weak initial noise ($\eta = 10^{-3}$) as average over 11 realisations. Sloped dashed lines indicate theoretical EKE growth with growths rates diagnosed from reference simulations with initial perturbations of specified wave numbers 3 and 4, respectively. Vertical dotted lines indicate transitions between stages characterised by different types of wave-wave interaction.

with non-linear triad interactions of certain wave modes with the dominant mode of the system once it reached maximum EKE. The continued growth of selected zonal wave modes seen in Fig. 6 eventually leads to a change in dominant wave number of the system (at about day 22 for both initial states). At this point the EKE of initially dominant wave mode 6 decreases and a succession of peaks in corresponding lower wave numbers occurs, starting with modes 4 and 2. This upscale energy cascade

is especially pronounced in the LC2 setup and consistent with the changes in PV distribution and total EKE related to a phase of noise-induced wave breaking (see Figs. 2 and 3). In particular, the consecutive occurrence of phases with cyclonic and anticyclonic wave breaking in the LC2 case is consistent with a shift of the system towards lower wave numbers over time (e.g., Hartmann and Zuercher, 1998; Orlanski, 2003).

## 4   Life-cycles with strong initial noise ($\eta \gg 1$)

In the previous section we found various changes in the evolution of LC1 and LC2 initialisations when the initial perturbation contains relatively weak noise but is still dominated by a single specified wave number ($\eta < 1$). This section discusses the extreme case of essentially random initial conditions without any preferred zonal dependence ($\eta \gg 1$).

Figure 7 shows the EKE evolution of LC1 and LC2 experiments in this strong-noise regime. The system generally follows the typical life-cycle behaviour with distinct wave growth and decay phases and an essentially steady final state. The initial wave

growth is strongest for wave numbers 7 and 8 in both settings (cf. Fig. S1 in the Supplement) with corresponding EKE peaks at about day 10 and decay phases afterwards. Subsequently lower wave numbers show peaks in EKE suggesting an upscale energy transfer which reaches wave number 1 about one to two weeks later. Fluctuations in individual wave numbers appear as





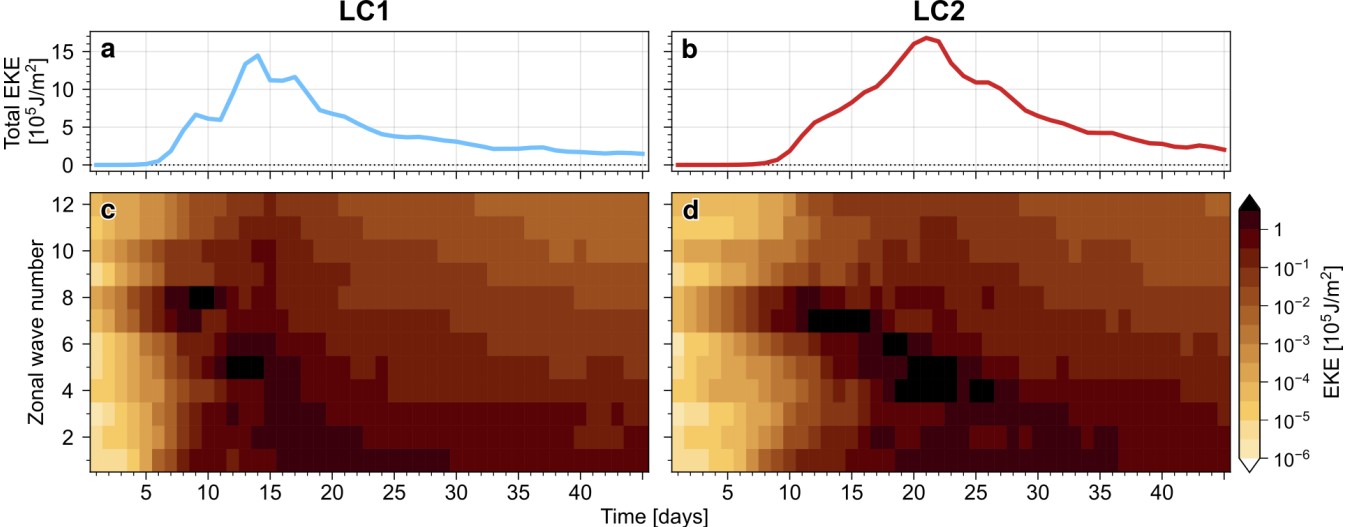

**Figure 7.** Evolution of EKE in experiments with LC1 (left column) and LC2 (right column) initial conditions and strong initial noise ($\eta = 10^3$). The top row shows the total EKE, the bottom row shows EKE for individual wave numbers; both as mean over three different noise realisations. Energies are shown as northern-hemispheric horizontally averaged and vertically integrated energy densities.

isolated maxima in the total EKE evolution (e.g., at day 8 for LC1 experiments). This upscale energy cascade is consistent with wave-wave interactions that are promoted by noise-induced secondary wave breaking also found in experiments with weak

initial noise (see Fig. 6).

The evolution of the PV structures is overall similar for the two strong-noise setups with LC1 and LC2 initial states (Fig. 8). Early cyclonic wave breaking dominated by higher wave numbers is followed by anticyclonic breaking at predominantly lower wave numbers. The transition between these two phases of opposite wave breaking is essentially seamless and consistent with our findings in terms of energetics (Fig. 7). Contrasting experiments with weak noise, however, the primary wave breaking is

entirely cyclonic in LC1 initialisations with strong noise. Despite the similarities in wave breaking, the final state PV distribution shows small differences between the LC1 and LC2 cases. While in both settings the final jet and the associated maximum in meridional PV gradient is located at about 60° latitude, we find a zone of reduced PV gradients equatorward of this jet for LC1 initialisations whereas such a "surf zone" (McIntyre and Palmer, 1983) is found poleward of the jet for LC2 initialisations.

The formation of these surf-zones on the equatorward or poleward flank of the jet suggest a tendency for slightly more

efficient mixing on the corresponding side due to anticyclonic and cyclonic wave breaking in LC1 and LC2 states, respectively, consistent with the paradigm of monochromatically perturbed baroclinic-wave life-cycles (and found for small initial noise strengths).

The consecutive occurrence of phases with cyclonic and anticyclonic wave breaking for experiments with strong initial noise ($\eta \gg 1$) is also reflected in the corresponding eddy momentum fluxes (Fig. 9). Both LC1 and LC2 experiments show pro-

nounced phases of negative momentum fluxes that subsequently turn positive, resulting in a net-positive momentum flux over

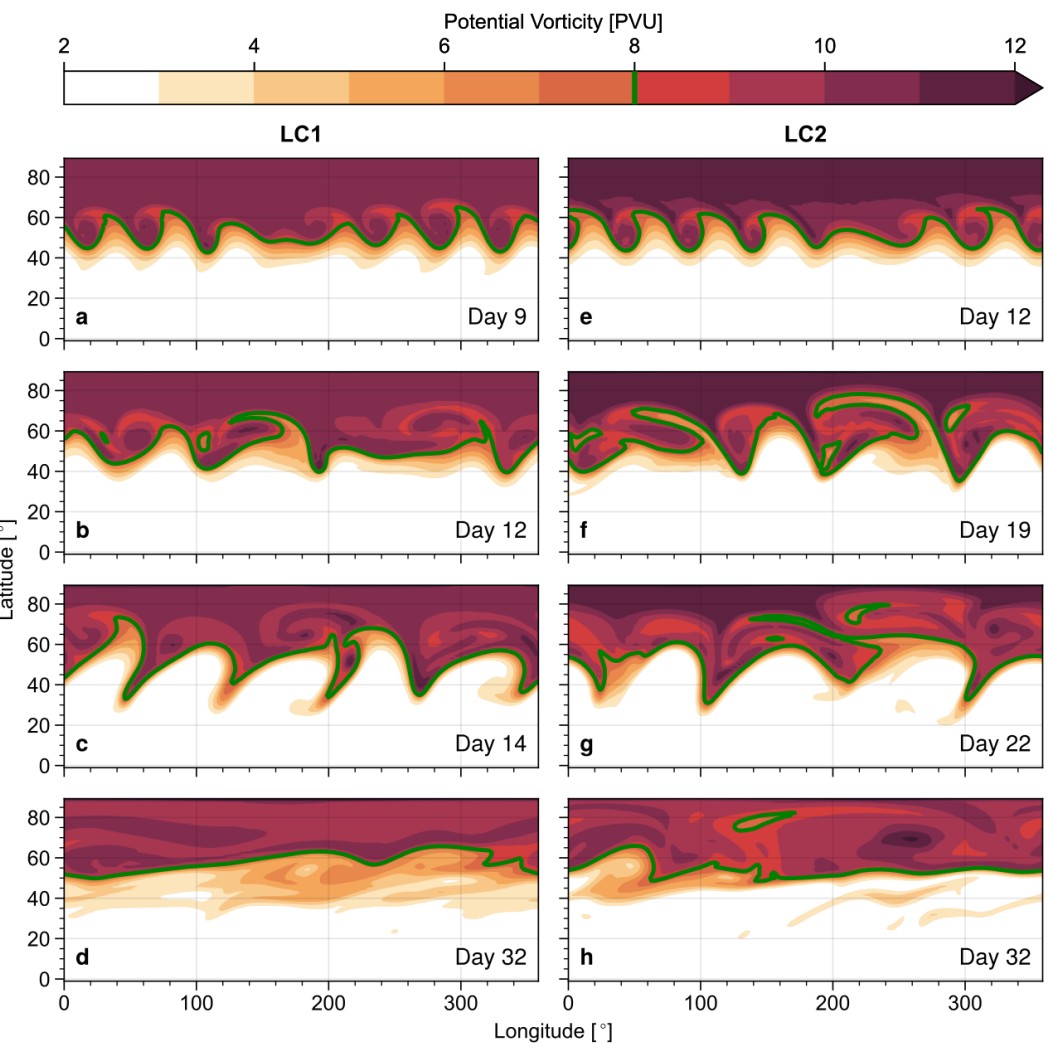

**Figure 8.** Potential vorticity on the 350 K isentrope at different days for LC1 (a-d) and LC2 (e-h) initial conditions and strong initial noise perturbations ($\eta = 10^3$). The 8 PVU contour is highlighted as visual aid.



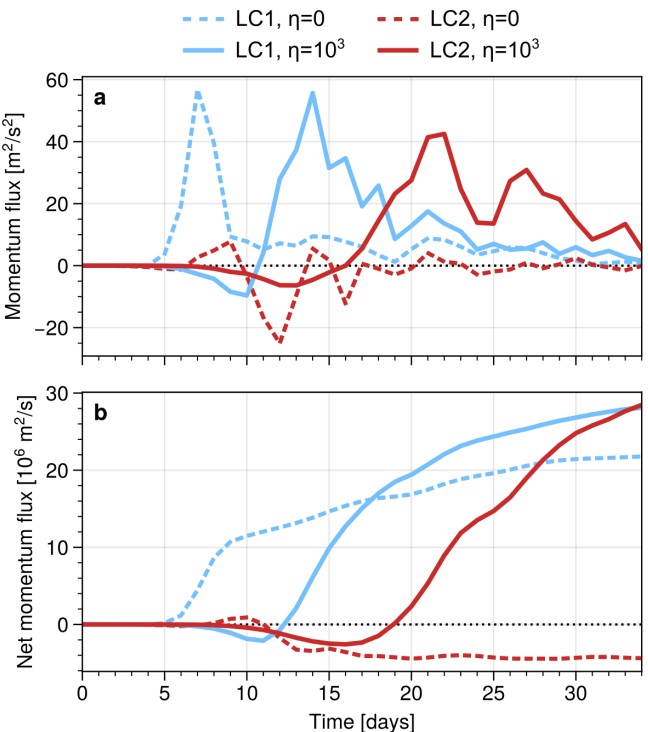

**Figure 9.** Evolution of (a) hemispherically averaged eddy momentum flux and (b) accumulated net eddy momentum flux for experiments with either LC1 or LC2 initial conditions and either no ($\eta = 0$) or strong ($\eta = 10^3$) initial noise perturbation. Experiments with noise show the average over three different realisations.

the course of the simulation. In contrast, experiments with monochromatic initial perturbation ($\eta = 0$) show a clear distinction with predominantly single-signed momentum fluxes and oppositely signed net-fluxes for LC1 and LC2 initial states. Figures 7, 8 and 9 show a pronounced shift of baroclinic-wave life-cycle characteristics towards a robust evolution for both initial states when using noisy initial perturbations: a clear net-poleward shift of the jet is observed during the life-cycle, while periods with equatorward shifted jet are purely transient.

## 5 Discussion

We found the evolution of baroclinic-wave life-cycles to be strongly modified when the initial conditions contain random noise perturbations rather than monochromatic perturbations with a single zonal wave number. In general life-cycles with noisy initial perturbation ($\eta \gg 1$) show a robust evolution over the course of the life-cycle, with both LC1 and LC2 experiments experiencing phases of cyclonic and subsequent anticyclonic wave breaking (Fig. 8). Associated with the direction in wave breaking during the corresponding phases are equatorward and poleward eddy momentum fluxes (Fig. 9), respectively. Hence





the life-cycles typically experience a transient phase with equatorward jet shift, but eventually, a poleward net-shift of the jet irrespective of the initial state (Fig. 5). Experiments with different realisations of initial noise perturbations indicate these findings to be robust.

This dominance of anticyclonic wave breaking and poleward momentum fluxes during all non-monochromatic life-cycles offers conceptual support for the observed behaviour in the real atmosphere, namely to favour anticyclonic wave breaking and overall net-poleward jet shifts periods (Feldstein, 1998; Lee et al., 2007; Birner et al., 2013). The generally reduced sensitivity of the life-cycle evolution to characteristics of the basic state for non-monochromatic initialisations further has potential implications for other baroclinic-wave life-cycle research. The dependency of life-cycle flavour on certain characteristics of

the system (Shapiro et al., 2001; Wittman et al., 2007) or the dependency of flow behaviour on the direction of wave breaking (Polvani and Esler, 2007; Wang and Polvani, 2011) could potentially be reduced in cases with noisy initial states.

Although the overall evolution of life-cycles with noisy initialisations is robust and insensitive to changes in the initial conditions (LC1 vs. LC2), certain characteristics stay distinctly different between initialisations. This includes the formation of "surf-zones" at the equatorward (LC1) or poleward (LC2) flanks of the final jet (Figs.3d and h), associated with overall

favouring of anticyclonic or cyclonic wave breaking during life-cycles (e.g., Fig. 9b). LC1 and LC2 initialisations further show persistent differences in terms of growth and decay rates of EKE (Fig. 7) for all magnitudes of initial noise. However, it is not clear if this should be interpreted as distinct characteristic of two qualitatively different types of life-cycle or a result of a quantitative change related to the structure of the respective basic state (and hence, e.g., changes in wave propagation properties).

Figures 6 and 7 suggest a substantial importance of non-linear interaction during the life-cycle. In particular, we find an upscale energy cascade via wave-wave interactions that leads to consecutive breaking of successively lower wave numbers. In experiments with relatively weak noise this wave-wave interaction is highly selective, transferring energy from the dominant mode to lower wave numbers that satisfy the fundamental wave number restrictions for triad-interactions. In the present case the dominant mode is given by wave number 6, due to the specified initial perturbation, and hence wave mode combinations

whose wave numbers sum up to 6 can efficiently extract energy from wave 6 and experience particularly accelerated growth. As an example, mode 6 can exchange energy with modes 2 and 4 via the triad interaction (2,4,6), but can also exchange energy with modes 1 and 5 via (1,5,6). The fact that Fig. 6 suggests asymmetries in the rate that energy is transferred to, e.g., modes 4 and 5 could be explained via the numeric argument that there potentially is a larger number of combination chains to transfer energy from the dominant mode 6 to mode 4 than to mode 5. In that sense interactions following the interaction chain (6,6,12),

(4,8,12) and (4,4,8) can effectively transfer energy from mode 6 to mode 4 via modes 8 and 12. We performed sensitivity experiments (not shown) forced by noisy initial perturbations with dominant wave number 7 ($\eta \approx 1$) and found a consistent accelerated growth of waves modes 2 and 5. (In principle, a full analysis of wave-wave interactions would take into account the total horizontal wave number, however we find the simple treatment in terms of zonal wave numbers only to provide useful insights into the system's behaviour in this case.)



## 6 Summary and conclusions

Our results show that the evolution of baroclinic-wave life-cycles is strongly modified when the initial state includes random noise perturbations compared to traditional monochromatic initial perturbations with a single zonal wave number, especially during the non-linear stages. If the initial noise is weak ($\eta \ll 1$) the early evolution of life-cycles with LC1 and LC2 initial states is qualitatively distinct and follows the well-known paradigms (Thorncroft et al., 1993). However, a phase of noise-induced secondary wave growth and decay occurs during the later stages of the life-cycle for both initial configurations, leading to substantial changes in energetics (Fig. 2) and PV structure (Fig. 3) of the corresponding final state.

While we find this noise-induced wave breaking to have a relatively weak effect on LC1 experiments, it leads to a complete breakdown of the typical LC2 final state, which is usually characterised by high levels of EKE and associated coherent cyclonic vortices. For both initial configurations the noise-induced wave breaking is predominantly anticyclonic and corresponds to strong poleward eddy momentum fluxes, resulting in a net-poleward jet shift during the life-cycle for LC1 and LC2 initialisations for all noise magnitudes. If the initial perturbation is noise dominated ($\eta \gg 1$), periods of equatorward jet shift occur irrespective of the initial conditions, but are purely of transient nature. Our analysis further shows that an upscale energy cascade via non-linear wave-wave interactions plays an important role in non-monochromatic life-cycles, transferring energy from synoptic to planetary wave numbers.

The findings presented in this study indicate that the dichotomy of LC1 and LC2 has reduced relevance in non-monochromatic conditions; they support that baroclinic mixing of air masses in the mid-latitudes dominantly induces poleward jet shifts.

## Appendix A: Construction of initial state

The initial conditions used in this study are defined via a zonally symmetric zonal wind field $U = U_j + U_s$, consisting of two individual components: a meridionally and vertically confined jet component $U_j$ defined via Equation A1 and a meridional wind shear component $U_s$ defined via Equation A2:

$$U_j = \hat{U}_j \left(z/z_j\right) \exp\left((1 - (z/z_j)^\alpha)/\alpha\right) \sin^3\left[\pi \sin^2(\phi)\right], \tag{A1}$$

$$U_s = \hat{U}_s \left(\exp\left[-(z/z_s)^2\right]\right) \left[\exp(-((\phi - \phi_j + \phi_s)/\Delta\phi_s)^2) - \exp(-((\phi - \phi_j - \phi_s)/\Delta\phi_s)^2)\right], \tag{A2}$$

where $z = -H \ln(p/p_0)$ is a log-pressure coordinate with scale height $H = 7.5$ km and reference pressure $p_0 = 1000$ hPa and $\phi$ describes latitude. For the jet profile the parameters $U_j$, $z_j$ and $\alpha$ can be used to modify the jet strength, the core height and the depth of the jet, respectively. For the meridional shear profile the parameters $\hat{U}_s$, $z_s$ define the strength of the shear, respectively, while $\phi_s$ and $\Delta\phi_s$ can be used to control its meridional width. The two initial states used in the present study only differ in terms of the parameter choice for $\hat{U}_s$, with LC1 corresponding to no shear ($\hat{U}_s = 0$) and LC2 corresponding to cyclonic shear ($\hat{U}_s = 10$ m/s); see Fig. 1. Sensitivity experiments (not shown) indicate that our results are robust over a range of different values for $\hat{U}_s$ and that an abrupt transition between predominantly LC1 and LC2 characteristics occurs at $\hat{U}_s \approx 7$ m/s in monochromatically perturbed experiments. Further, using a barotropic or baroclinic meridional shear profile did not lead to





major differences in life-cycle behaviour (consistent with the findings of Hartmann, 2000). Note that we confine the jet profile to the northern hemisphere, i.e., we choose $U \equiv 0$ for $\phi < 0$ and therefore keep the southern hemisphere initially at rest. Table A1 lists the physical parameters used to define the different initial conditions used in all experiments.

**Table A1.** Physical parameters used in the the different model experiments.

| Symbol | Physical meaning | Value |
|:---:|:---:|:---:|
| $\hat{U}_j$ | Jet strength | 45 m/s |
| $z_j$ | Jet core height | 11 km |
| $\alpha$ | Jet depth parameter | 3 |
| $\hat{U}_s$ | Shear strength | 0 and 10 m/s |
| $z_s$ | Shear depth | 9 km |
| $\phi_s$ | Meridional width of the shear | 15° |
| $\Delta\phi_s$ | Sharpness of the shear | 12.5° |

From the initial wind field we compute the meridionally varying part of the initial temperature field following the thermal wind balance approach used by Polvani and Esler (2007). The meridionally constant part of the (potential) temperature field $T(z)$ is chosen to follow a piecewise linear profile with surface temperature $T(0) = 288.15$ K and vertical gradient specified via Table A2, motivated by the US standard atmosphere (NOAA, 1976). To improve the numerical stability due to sharp transitions in $\partial T(z)/\partial z$ at the turning points we applied a 2 km running mean to the temperature profile post construction. Note that our numerical model (see Sec. 2) uses a surface of constant pressure as lower boundary and hence does not require any balancing adjustment of the surface pressure field.

**Table A2.** Piecewise constant vertical gradients used to define the horizontally constant temperature profile $T(z)$.

| Height range [km] | Temperature gradient $\partial T/\partial z$ [K/km] |
|:---:|:---:|
| $0 < z < 11$ | -6.5 |
| $11 < z < 20$ | 0 |
| $20 < z < 30$ | 1 |
| $30 < z < 45$ | 2.8 |
| $45 < z < 50$ | 0 |
| $50 < z < 75$ | -2.8 |
| $75 < z$ | 0 |

In all experiments wave growth on the baroclinically unstable initial jet is triggered via a small temperature perturbation $\tilde{T}(z, \phi, \lambda) = \tilde{T}_z(z)\tilde{T}_\phi(\phi)\left[\tilde{T}_w(\lambda) + \tilde{T}_n(\lambda)\right]$, where $\tilde{T}_w$ and $\tilde{T}_n$ describe two different components of zonal dependency.





$\tilde{T}_w = T_w \cos(k\lambda)$ describes a wave perturbation in longitude ($\lambda$) with a single specified zonal wave number $k$ and amplitude $T_w$, while $\tilde{T}_n$ describes a noise perturbation defined via zonally uncorrelated random noise with underlying uniform proba-
bility distribution spanning $[-T_n, T_n]$, where $T_n$ specifies the noise amplitude. The vertical and meridional structure of the perturbation are given via

$$\tilde{T}_z = \exp\left[(p - p_0)/(p_0 - p_{pert})\right], \tag{A3}$$
$$\tilde{T}_\phi = \cosh^{-2}\left[2\left(\phi - \phi_{pert}\right)\right], \tag{A4}$$

where $p_0 = 1000$ hPa, $p_{pert} = 700$ hPa and $\phi_{pert} = 45°$.

*Acknowledgements.* This work is resulted from the master project of Felix Jäger under supervision of Philip Rupp and Thomas Birner at the Meteorological Institute Munich, which provided the computational infrastructure for the simulations and their analysis. This research has further been supported by the German Research Foundation (DFG) (grant no. SFB/TRR 165; Waves to Weather project). We also want to thank Andrew Charlton-Perez for very helpful and inspirational comments on the results of this study.

*Author contributions.* FJ conducted the numerical experiments, produced the figures, performed the main part of the analysis and composed the first draft of this manuscript. PR assisted with the numerical and conceptual experiment design, contributed to the analysis and interpretation of the results and revised parts of the manuscript. TB advised FJ throughout his work, contributed to the analysis and interpretation of the results and improved the paper for the final version.

*Code availability.* A detailed description of the model setup and the idealised experimental design is given in Section 2
.

*Data availability.* No data sets were used in this article.

*Competing interests.* At least one of the (co-)authors is a member of the editorial board of Weather and Climate Dynamics.



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
