# Peer review of "Robust poleward jet shifts in idealised baroclinic-wave life-cycle experiments with noisy initial conditions"

_Weather and Climate Dynamics, 2022_

## Author Comment (AC1)

We thank Dennis Hartmann for carefully reading our manuscript, and for his constructive comments. In the following we will respond to the various comments and point out any changes we intend to make to the paper based on them. Note that we have not provided exact manuscript corrections at this point, but we have provided the outline of planned changes. Line numbers and figure references in the reviewer's comments refer to the original manuscript. The reviewer's comments are in *black italics;* our responses are in blue.

*This paper is an interesting contribution to the literature on the impact of baroclinic shear on baroclinic lifecycles. Rather than using a single wavenumber to initialize the experiments, the authors add varying degrees of spatial white noise to the initial state. In cases of weak noise longer wavelengths grow via wave-wave interactions, (2,4,6) in the case of a base wavenumber of 6. These longer wavelengths are able to propagate toward the equator and lead to net poleward momentum flux in the case with large cyclonic barotropic shear (LC2) case as well as the LC1 case without the added cyclonic barotropic shear (LC1). If a high level of noise is added, shorter wavelengths, which are presumably more linearly unstable than wave 6, also develop early in the simulation and appear to break poleward in both the LC1 and LC2 cases. This leads to a situation where an initial stage of poleward wave breaking always occurs, but is always followed by equatorward wave propagation and breaking as the energy cascades to longer wavelengths that can propagate across the barotropic shear to the tropics. This leads one to conclude that equatorward wave propagation, poleward momentum flux, and poleward jet propagation must be a dominant feature of the general circulation, as is required by the global angular momentum balance*
.
Thank you for these encouraging summary remarks.

*Figure 3 , panels c and g are chosen at a particular time when wavenumber 4 dominates the image of PV. This misled me into thinking that wave 4 was growing by linear instability, which is not the thesis of the paper. Looking at Fig. 6 it is more obvious that this particular time is special. It would be good to note at this point that wave 2 is also evident in Fig. 3g or make some other comments to say that the dominance of wave 4 at this time is just transitory.*

Thank you for pointing this out. We indeed conclude that wave numbers 2 and 4 grow mostly due to non-linear interaction and not due to a simple linear instability. One argument here would also be that the initial state is much more unstable for other wavenumbers (like 5 and 7) than wavenumbers 4 and 2 (see Fig. S1 in the supplement).
Clarification in the text will be added. See also answers to comments on Fig. 3 and line 138 below.

*This is an interesting contribution and is fairly clearly written, with some exceptions that are noted below on a line-by-line and figure basis.*

*Comments on text:*

*Line 99: 'gradually'*
Will be done.

*115: Not sure what is meant by the initial phrase "Consistent with the energetics of the systems, "*
We meant to express the consistency between the evolution of EKE and MKE with the PV dynamics during the additional noise-induced cycle. In the text we will adapt the sentence to be clearer on that.

*117: Would a linear analysis of the zonal mean state at this time reveal that the most unstable wavenumber is 4? Is the energy of wave 4 coming from the mean state or WMF interactions?*
This is an nice idea for potentially gaining deeper understanding of how wave 4 and 2 grow. However, we have conducted experiments on different levels of noisiness with resulting timings of secondary wave growth (comp. e.g. discussion of Fig. 5 in original manuscript). We found growth rates above the ones that wave 4 and 2 would follow if they would grow purely due to linear instability. Additionally such accelerated growth during different stages of the wave 6 cycle suggests that a sufficiently large amplitude of wave 6 is necessary for the growth of other waves independent of the current zonal mean state. Further, the applicability of linear stability theory is likely limited given the highly non-linear nature of the wave-breaking phase.

*Fig. 3 in both cases, wavenumber 4 emerges as dominant around day 22-24. Why? It would be good at this point to say that you have picked out a particular time when wave 4 was dominant, and also point out that wavenumber 2 can also be seen at this time in panels C and G. The choice of time makes it look like it is mostly linear growth of wave 4, which is not consistent with the nonlinear theory that is actually the thesis of the paper.*
Clarification in the text will be added (planned to be inserted in line 117).

*135: Is that because wavenumber 4 (and 2) can propagate toward the equator, while wavenumber 6 cannot in the LC2 state?*
Thank you for this suggestion. Indeed, in our set up, wave numbers k<6 seem to be able to propagate equatorwards more easily than k=6 in the LC2 state. When wave 6 breaks, the LC2 state leads to a poleward wave-activity flux (comp. Fig. AC1 top right). However during the second wave breaking, where wave 4 and 2 dominate, the wave activity flux points towards the equator in both the LC1 and the LC2 setting. This can be also seen for strong noise in Fig. AC2 in both panels (LC1 and LC2) during the wave breaking of wave 4. Wave activity flux is equatorwards in both settings. We plan to include a comment on this.

[Figure]

Fig. AC1: Eliassen-Palm flux for weak noise runs of LC1 (left) and LC2 (right) during the first (top) and the second wave breaking (bottom) indicated by arrows. Its horizontal component additionally is shown with the shading.

[Figure]

Fig. AC2: Eliassen-Palm flux indicated by arrows for strong noise runs of LC1 (left) and LC2 (right) during the five days around the EKE maximum, i.e. during the wave breaking of wave 4. Its horizontal component additionally is shown with the shading.

*138: On first reading, I did not quite get the physical reason for the emergence of wavenumber 4, which seems to be key. I don't see any reason for a state consisting of wavenumber 0 and 6 to create wavenumber 4 through nonlinear exchange, but if I look back at Fig. 3 panel G, I can see some wavenumber 2. It might help to point that out. Wavenumber 4 can propagate toward the equator and produce an LC1 outcome in the end.*

This will also be covered by the insertion in line 117 indicated above.

*174: If the wave breaking event creates a spectrum of wavenumbers, why is the initial noise so important to the evolution of the flow after the first wave-breaking phase?*

The non-linear triad interactions during the primary wave breaking event in monochromatically perturbed experiments only project on multiples of the perturbation wave number, i.e., in our case creates waves with wave numbers 6, 12, 18, 24, etc.. Some authors even limited their model to have a strict wave-6 symmetry (comp. Magnusdottir, G. and Haynes, 1996). Other wave numbers only become important because initialized as noise. Our understanding is that once wave 6 has become sufficiently strong, these wave numbers extract energy from wave 6 to grow faster than expected based on linear baroclinic instability theory.

*Fig. 6 The legend " Specified wave 4" Is unclear. The other experiment was Specified wave 6, but it was allowed to evolve nonlinearly, whereas the curves for 4 and 3 seem to be extrapolations of their infinitesimal linear growth rates.*

We will try to further clarify this in legend and caption. The dashed lines in Fig. 6 indicate an evolution with the linear growth rate of wave number k estimated via a linear fit of EKE during the initial growth phase of an experiment initialised with a single wave number k perturbation and no noise.

*Fig. 6 If it is nonlinear wave exchange responsible for the growth of 2 and 4, why is their growth rate independent of the amplitude of wave 6? Their growth looks exponential, like they were linearly unstable.*

Indeed, the growth of wave 4 and 2 is exponential, however with rates that seem to react to the amplitude of the driving wave 6. As mentioned in lines 168-171 in the original manuscript, the rates of k=1,2,3,4,5, which lie above the ones seen for linear instability in the reference runs, start to diverge in LC1 around day 8 and in LC2 around day 11. Selective triad interactions enhance the growth of waves 2 and 4 more than waves 1,3,5. In both experiments, this correlates with the peak in wave 6 EKE. Drops in growth rate correlate with a drop in wave 6 EKE. We interpret the growth of the noise, in particular wave 4 and 2, to be a combination of their own instability and accelerated growth via non-linear energy transfer.

*194: Did you mean to say, "In contrast to experiments with weak noise," As it is, itconfused me. So in a case with white noise initialization, shorter wavelengths grow faster and tend to exhibit LC2 initial evolution, until the larger scales develop, which are able to propagate toward the equator, ending in a poleward jet shift and a more LC1-like final state.*

We agree with your comment. The wording of the sentence will be adapted.

*265: One might imagine a region of parameter space where the baroclinic growth of shorter wavelengths would be fast compared to the cascade to longer wavelengths in which the cyclonic state could be maintained by the poleward breaking of these shorter*

*waves. It might also be possible that the shorter waves contribute their energy to a stationary wave, such as in the blocking ridge situation.*

This is an intriguing idea. It would be very interesting for potential future work to explore this parameter space. In some way, the monochromatically and weakly perturbed LC2 experiments (Figs. 2 and 3) show this behaviour. There seems to be a threshold wave number $k_0$ above which we observe LC2 behaviour and below which we observe LC1 behaviour. We find high wave numbers to grow fastest in these experiments and lead to (quasi-)stable standing wave patterns (before short wave numbers start to dominate in cases with eta>0).

*Clearly for the general circulation to work, the dominant direction of eddy propagation and breaking must be toward the equator to satisfy the angular momentum balance.*

Thanks for stating clearly the consistency of our findings with this fundamental principle. We agree that given that surface winds are easterly/westerly between low/high latitudes this is a nice heuristic argument for our core results. However, the dominance of equatorward breaking does not by itself preclude additional modifications due to poleward breaking, as long as the latter is weaker than the former (as is the case in the real atmosphere). The heuristic argument cannot answer the question of whether the total average is comprised of quasi-steady LC1 and LC2 states as described by Thorncroft et al. (1993) (with LC1 anomalies being stronger), or whether one of these exists as a purely transitory phenomenon (as implied by our results). We plan to include a related comment.

---

## Author Comment (AC2)

We thank the reviewer for carefully reading our manuscript, and for her constructive comments. In the following we will respond to the various comments and point out any changes we intend to make to the paper based on them. Note that we have not provided exact manuscript corrections at this point, but we have provided the outline of planned changes. Line numbers and figure references in the reviewer's comments refer to the original manuscript. The reviewer's comments are in *black italics;* our responses are in blue.

*Eddy life cycle experiments have been used as a framework for understanding eddy-mean flow interactions in the midlatitude atmosphere for decades, as highlighted by the references provided by the authors and a recent review (Maher et al. 2019). In this study, the authors show that the sensitivity of the final jet state to the initial jet state may partly be an artifact of the idealized nature of traditional eddy life cycle experiments. When a single wavenumber is forced, wave breaking is very sensitive to meridional shear: with low shear, waves break anticyclonically, shifting the jet poleward (LC1), while with higher shear, waves tend to break cyclonically, shifting the jet equatorward (LC2).*

*The authors consider variations on these single wave, or monochromatic, experiments by adding noise of varying levels to excite all wavenumbers. They show that even in the limit of very weak noise, the the final state of LC1 and LC2 lifecycles are quite similar due to secondary wave breaking that occurs after the initial anticyclonic or cyclonic breaking event. The net change is primarily to LC2 cycle, where the second breaking event is anticyclonic, shifting the jet back poleward. Thus the shear has a large impact on the initial wave breaking event, but less so on the final state.*

*I think these are interesting results which merit publication after the authors consider the following minor revisions. It is remarkable that we are still learning about lifecycle experiments after almost half a century!*
Thank you for these encouraging summary remarks.

*General comment*
*Throughout much of the paper I was concerned about how the results depend on the initial noise. This is to say, with a different realization of the noise, could the evolution of the lifecycle be materially different? This concern was partially addressed by results from 3 member ensembles (in the discussion surrounding Figure 7), but even here, it's not possible to gauge the variance across the ensemble. I take it that the lifecycles proceed more or less the same way as long as their is some noise in the relevant wave numbers (waves 1-10 or so); even if the most important wavenumber for the secondary breaking event (wave 4) was weakly forced by the noise, nonlinear transfer of energy would invigorate it. But it would be good to establish this early in the paper.*
Thanks for pointing out. Indeed we will state clearer how our results depend on the noise realization we use. Based on sensitivity experiment with different realizations we found our results to be mostly independent of the initialized wave spectrum. Experiments in which the initial random noise perturbation only projected weakly on wavenumbers 2 and 4 showed the same qualitative behaviour as experiments which strong wavenumber 2 and 4 contributions in their initial noise. This further suggests the importance of the scale-selective non-linear interaction which accelerates the growth of waves 2 and 4 via energy transfer from the dominant wave 6.

*To be constructive, would it be possible to show a few additional experiments (initiated with different noise) in Figure 2. (And possibly Figure 4, which shows the final jet states for the same integrations.) I hope that additional solid lines for the \eta=10^-3 experiments would not overly crowd the figure. If all the low noise experiments look exactly the same, the authors could just state this in the text and alleviate my concern from the start.*

We plan to provide a figure on the evolution of the ensemble members illustrating the small intra-ensemble variability compared to the difference of the evolutions.

*Another option would be an additional figure showing that the evolution of key quantities (momentum fluxes, EKE, etc.) follow very similar trajectories for different initializations of noise for all levels of \eta. (Perhaps the variation in noise matter more when \eta is large?) The key is to establish that the difference between lifecycles with different noise realizations is small compared to the difference between the experiments with noise and the monochromatic experiments.*

We agree, this key question will be adressed in detail with a figure in the supplement as stated above.

*Minor comments by wavenumber*

*12-3. I found this line to be a bit awkward. Consider "… for LC2 initialisations are found to become unstable eventually, with the onset of instability coming sooner for larger noise perturbations."*

Line will be adapted.

*28 "flavours, or paradigms, of"*

Will be done.

*Paragraph at 66: As the noise is the major contribution of the manuscript, it might be nice to explain the gist of it in the text. For instance, you could say that the perturbations are white in space, equally exciting all wavenumbers (on average). Perhaps this could be done at line 77 where the amplitude of the noise is introduced.*

Will be done.

*74 along the same lines, could you briefly characterize the meaning of parameter \hat{U_s} in the text, referring to the equation number in the appendix.*

Will be done.

*77 Appendix90. It was around here that I started worrying whether the realization of the initial noise*
*mattered to the lifecycle. If it does not, a sentence here could put the reader at ease. This could also be discussed in the figure caption.*

Additional to a supplemental figure, the sentence in lines 82-83 will be slightly changed to make this clearer from start.

*107 This is just a comment about style, but I find that footnotes almost over emphasize the point, as the reader breaks off the text to get to it. Consider just putting this material in the main text.*
Thanks for that suggestion, we will put this footnote into the main text.

*138. Could you describe this noise induced wave breaking as a secondary instability? The flow is presumably now stable to wave 6 perturbations, but not others?*
Indeed, the growth of the noise could potentially be due to a secondary instability. However, there are several indicators that the increase in EKE for waves 4 and 2 for a second wave breaking is heavily fostered by wave 6 and only partially can be explained by linear theory. One indication for non-linear interactions between the different zonal wave numbers is the deviation in growth rate from the prediction of linear theory, which is already visible during the first days of the simulation. Furthermore, the EKE of wave 6 drops, when wave 4 and 2 reach substantial amplitudes (see e.g. Fig. 6b, day 22). This seems to point to an energy flux from 6 to 4 and 2. Nonlinear processes appear to alter the growth of the noise significantly in all non-monochromatic simulations.

*142-145. This line seemed to come too early in the text. Please shift it back after Figure 5a is introduced and the result has been established.*
Will be done.

*167-7. Is this really similar to quasi-linear non-normal growth? That process is rather distinct from nonlinear wave interactions. Please provide more evidence to support this statement.*
Thank you for pointing this out. We tried to express that the observed growth cannot be explained by single normal mode growth. We do not intend to focus on the distinction of normal vs. non-normal growth, but rather on the non-linear mechanisms still resulting in accelerated, quasi-linear growth. We will adapt the passage accordingly.

*174. Is "upscale energy cascade" an appropriate way to describe this? Consider "upscale energy transfer" as the flow does not appear to be fully turbulent.*
We agree, the substitution of "cascade" with "transfer" will be done.

*213. What is "In general" meant to signify here. Is this is reference to the fact that different realizations of noise can lead to different behavior? Or does it refer to difference that occur with variations in the shear parameter \hat{U}_s or other qualities of the initial jet state.*
We will include a clearer wording.

*218. Same for "typically".*
We will drop this word.

*222 Could you clarify what is meant by "overall net-poleward jet shift periods." The tendency of anticyclonic breaking to shift the jet poleward should be reflected in the mean state.*

"periods" will be removed. Then it says "overall net-poleward jet shifts"

*232-4 The meaning of this sentence was a bit obscure to me. Do the authors mean that the remarkably sensitivity of monochromatic experiments to \hat{U}_s, which justified the LC1 vs LC2 paradigms, may not be justified with noise? That is, there aren't really two kinds of wave cycles, but rather a continuum?*

Indeed, describing whole simulations of a longer period including several wave breaking events in different directions with the LC1-LC2 dichotomy might not be justified. For single wave breaking events or phases however, we still deem the paradigms to be useful. In some experiments with only noise as initial perturbation, we even found spatial differences in wave breaking direction at certain times (LC1-like behaviour at some longitudes and LC2-like behaviour at others). We will try to make these aspects clearer.

*236 and 263. Again, consider "upscale energy transfer"*

Will be done.

---

## Author Response (AR1)

We thank the two referees for carefully reading our manuscript, and for their constructive comments. In the following we will respond to the various comments of each referee and point out any changes we made to the paper based on them. Line numbers and figure references in the reviewer's comments refer to the original manuscript, line and figure references in the responses refer to the revised version. The reviewer's comments are in *black italics;* our responses are in blue, the changes in the text are in *blue italics*.

**Response to Referee #1:**

*This paper is an interesting contribution to the literature on the impact of baroclinic shear on baroclinic lifecycles. Rather than using a single wavenumber to initialize the experiments, the authors add varying degrees of spatial white noise to the initial state. In cases of weak noise longer wavelengths grow via wave-wave interactions, (2,4,6) in the case of a base wavenumber of 6. These longer wavelengths are able to propagate toward the equator and lead to net poleward momentum flux in the case with large cyclonic barotropic shear (LC2) case as well as the LC1 case without the added cyclonic barotropic shear (LC1). If a high level of noise is added, shorter wavelengths, which are presumably more linearly unstable than wave 6, also develop early in the simulation and appear to break poleward in both the LC1 and LC2 cases. This leads to a situation where an initial stage of poleward wave breaking always occurs, but is always followed by equatorward wave propagation and breaking as the energy cascades to longer wavelengths that can propagate across the barotropic shear to the tropics. This leads one to conclude that equatorward wave propagation, poleward momentum flux, and poleward jet propagation must be a dominant feature of the general circulation, as is required by the global angular momentum balance.*

Thank you for these encouraging summary remarks.

*Figure 3 , panels c and g are chosen at a particular time when wavenumber 4 dominates the image of PV. This misled me into thinking that wave 4 was growing by linear instability, which is not the thesis of the paper. Looking at Fig. 6 it is more obvious that this particular time is special. It would be good to note at this point that wave 2 is also evident in Fig. 3g or make some other comments to say that the dominance of wave 4 at this time is just transitory.*

Thank you for pointing this out. We indeed conclude that wave numbers 2 and 4 grow mostly due to non-linear interaction and not due to a simple linear instability. One argument here would also be that the initial state is much more unstable for other wavenumbers (like 5 and 7) than wavenumbers 4 and 2 (see Fig. S1 in the supplement).
The following is added in lines 121-123:

*"The time picked for illustration in Fig. 3c and g highlights the dominance of wave 4. However, also wave 2 and other wave numbers contribute to the noise-induced anticyclonic wave breaking."*

See also answers to comments on Fig. 3 and line 138 below.

*This is an interesting contribution and is fairly clearly written, with some exceptions that are noted below on a line-by-line and figure basis.*

*Comments on text:*

*Line 99: 'gradually'*
Done.

*115: Not sure what is meant by the initial phrase "Consistent with the energetics of the systems, "*
We meant to express the consistency between the evolution of EKE and MKE with the PV dynamics during the additional noise-induced cycle. In the text we adapted the sentence to be clearer on that. We changed the wording to (lines 119-120):

*"Consistent with the evolution of the system in terms of EKE and MKE, an additional noise-induced cycle of pronounced wave growth and breaking occurs at about day 22 (Figs. 3c and g)."*

*117: Would a linear analysis of the zonal mean state at this time reveal that the most unstable wavenumber is 4? Is the energy of wave 4 coming from the mean state or WMF interactions?*
Thank you for this interesting suggestion to look at linear stability of the zonal mean state during different times of the life cycle. However, our existing results strongly point toward an important role of non-linear wave interactions during growth of wave 4 here. For example, we have conducted experiments on different levels of noisiness with resulting timings of secondary wave growth (comp. e.g. discussion of Fig. 5 in original manuscript). We found growth rates above the ones that wave 4 and 2 would follow if they would grow purely due to linear instability. Additionally, such accelerated growth during different stages of the wave 6 cycle suggests that a sufficiently large amplitude of wave 6 is necessary and sufficient condition for the growth of other waves independent of the current zonal mean state. Further, the applicability of linear stability theory is likely limited given the highly non-linear nature of the wave-breaking phase.
No changes to the text have been made.

*Fig. 3 in both cases, wavenumber 4 emerges as dominant around day 22-24. Why? It would be good at this point to say that you have picked out a particular time when wave 4 was dominant, and also point out that wavenumber 2 can also be seen at this time in panels C and G. The choice of time makes it look like it is mostly linear growth of wave 4, which is not consistent with the nonlinear theory that is actually the thesis of the paper.*
Clarification in the text added (inserted in lines 121-123).

*135: Is that because wavenumber 4 (and 2) can propagate toward the equator, while wavenumber 6 cannot in the LC2 state?*
Thank you for this suggestion. Indeed, in our set up, wave numbers k<6 seem to be able to propagate equatorwards more easily than k=6 in the LC2 state. When wave 6 breaks, the LC2 state leads to a poleward wave-activity flux (comp. Fig. AC1 top right). However during the second wave breaking, where wave 4 and 2 dominate, the wave activity flux points towards the equator in both the LC1 and the LC2 setting. This can be also seen for strong noise in Fig. AC2 in both panels (LC1 and LC2) during the wave breaking of wave 4. Wave activity flux is equatorwards in both settings. We included the following comment in line 249.

[Figure]

Fig. AC1: Eliassen-Palm flux for weak noise runs of LC1 (left) and LC2 (right) during the first (top) and the second wave breaking (bottom) indicated by arrows. Its horizontal component additionally is shown with the shading.

[Figure]

Fig. AC2: Eliassen-Palm flux indicated by arrows for strong noise runs of LC1 (left) and LC2 (right) during the five days around the EKE maximum, i.e. during the wave breaking of wave 4. Its horizontal component additionally is shown with the shading.

*"These longer waves appear to propagate more easily to the equator, resulting in anticyclonic breaking."*

*138: On first reading, I did not quite get the physical reason for the emergence of wavenumber 4, which seems to be key. I don't see any reason for a state consisting of wavenumber 0 and 6 to create wavenumber 4 through nonlinear exchange, but if I look back at Fig. 3 panel G, I can see some wavenumber 2. It might help to point that out. Wavenumber 4 can propagate toward the equator and produce an LC1 outcome in the end.*
We covered this with the insertion in lines 121-123 indicated above.

*174: If the wave breaking event creates a spectrum of wavenumbers, why is the initial noise so important to the evolution of the flow after the first wave-breaking phase?*
The non-linear triad interactions during the primary wave breaking event in monochromatically perturbed experiments tend to primarily project on multiples of the perturbation wave number, i.e., in our case create waves with wave numbers 6, 12, 18, 24, etc.. Some authors even limited their model to have a strict wave-6 symmetry (comp. Magnusdottir, G. and Haynes, 1996). In our noisy experiments the entire spectrum participates in the dynamics during all stages of the life cycle.
No changes to the text have been made.

*Fig. 6 The legend " Specified wave 4" Is unclear. The other experiment was Specified wave 6, but it was allowed to evolve nonlinearly, whereas the curves for 4 and 3 seem to be extrapolations of their infinitesimal linear growth rates.*
We further clarified this in legend and caption. The dashed lines in Fig. 6 indicate an evolution with the linear growth rate of wave number k estimated via a linear fit of EKE during the initial growth phase of an experiment initialised with a single wave number k perturbation and no noise.

*Fig. 6 If it is nonlinear wave exchange responsible for the growth of 2 and 4, why is their growth rate independent of the amplitude of wave 6? Their growth looks exponential, like they were linearly unstable.*
Indeed, the growth of wave 4 and 2 is exponential, however with rates that seem to react to the amplitude of the driving wave 6. As mentioned in lines 168-171 in the original manuscript, the rates of k=1,2,3,4,5, which lie above the ones seen for linear instability in the reference runs, start to diverge in LC1 around day 8 and in LC2 around day 11. Selective triad interactions enhance the growth of waves 2 and 4 more than waves 1,3,5. In both experiments, this correlates with the peak in wave 6 EKE. Drops in growth rate correlate with a drop in wave 6 EKE. We interpret the growth of the noise, in particular wave 4 and 2, to be a combination of their own instability and accelerated growth via non-linear energy transfer.
No changes to the text have been made.

*194: Did you mean to say, "In contrast to experiments with weak noise," As it is, it confused me. So in a case with white noise initialization, shorter wavelengths grow faster and tend to exhibit LC2 initial evolution, until the larger scales develop, which are able to propagate toward the equator, ending in a poleward jet shift and a more LC1-like final state.*
We agree with your comment. We adapted the wording of the sentence to *"In contrast to ..."*

*265: One might imagine a region of parameter space where the baroclinic growth of*

*shorter wavelengths would be fast compared to the cascade to longer wavelengths in which the cyclonic state could be maintained by the poleward breaking of these shorter waves. It might also be possible that the shorter waves contribute their energy to a stationary wave, such as in the blocking ridge situation.*

This is an intriguing idea. It would be very interesting for potential future work to explore this parameter space. In some way, the monochromatically and weakly perturbed LC2 experiments (Figs. 2 and 3) show this behaviour. There seems to be a threshold wave number $k_0$ above which we observe LC2 behaviour and below which we observe LC1 behaviour. We find high wave numbers to grow fastest in these experiments and lead to (quasi-)stable standing wave patterns (before short wave numbers start to dominate in cases with eta>0).

No changes to the text have been made.

*Clearly for the general circulation to work, the dominant direction of eddy propagation and breaking must be toward the equator to satisfy the angular momentum balance.*

Thanks for stating clearly the consistency of our findings with this fundamental principle. We agree that given that surface winds are easterly/westerly between low/high latitudes this is a nice heuristic argument for our core results. However, the dominance of equatorward breaking does not by itself preclude additional modifcations due to poleward breaking, as long as the latter is weaker than the former (as is the case in the real atmosphere). The heuristic argument cannot answer the question of wether the total average is comprised of quasi-steady LC1 and LC2 states as described by Thorncroft et al. (1993) (with LC1 anomalies being stronger), or whether one of these exists as a purely transitory phenomenon (as implied by our results). We included a related comment (lines 229-234):

*"This is consistent with the heuristic argument of global angular momentum balance requiring equatorward, anticyclonic wave breaking to dominate. However, this does not by itself preclude additional modifcations due to poleward breaking, as long as the latter is weaker than the former. The argument of angular momentum balance cannot answer the question of whether the total average is comprised of the kind of quasi-steady LC1 and LC2 states described by Thorncroft et al. (1993) (with LC1 anomalies being stronger), or whether LC2 exists as a purely transitory phenomenon (as implied by our results)."*

**Response to Referee #2:**

*Eddy life cycle experiments have been used as a framework for understanding eddy-mean flow interactions in the midlatitude atmosphere for decades, as highlighted by the references provided by the authors and a recent review (Maher et al. 2019). In this study, the authors show that the sensitivity of the final jet state to the initial jet state may partly be an artifact of the idealized nature of traditional eddy life cycle experiments. When a single wavenumber is forced, wave breaking is very sensitive to meridional shear: with low shear, waves break anticyclonically, shifting the jet poleward (LC1), while with higher shear, waves tend to break cyclonically, shifting the jet equatorward (LC2).*

*The authors consider variations on these single wave, or monochromatic, experiments by adding noise of varying levels to excite all wavenumbers. They show that even in the limit of very weak noise, the the final state of LC1 and LC2 lifecycles are quite similar due to secondary wave breaking that occurs after the initial anticyclonic or cyclonic breaking event. The net change is primarily to LC2 cycle, where the second breaking event is anticyclonic, shifting the jet back poleward. Thus the shear has a large impact on the initial wave breaking event, but less so on the final state.*

*I think these are interesting results which merit publication after the authors consider the following minor revisions. It is remarkable that we are still learning about lifecycle experiments after almost half a century!*
Thank you for these encouraging summary remarks.

*General comment*
*Throughout much of the paper I was concerned about how the results depend on the initial noise. This is to say, with a different realization of the noise, could the evolution of the lifecycle be materially different? This concern was partially addressed by results from 3 member ensembles (in the discussion surrounding Figure 7), but even here, it's not possible to gauge the variance across the ensemble. I take it that the lifecycles proceed more or less the same way as long as their is some noise in the relevant wave numbers (waves 1-10 or so); even if the most important wavenumber for the secondary breaking event (wave 4) was weakly forced by the noise, nonlinear transfer of energy would invigorate it. But it would be good to establish this early in the paper.*
Thanks for pointing out. We now state clearer how our results depend on the noise realization we use (lines 83-85). Based on sensitivity experiment with different realizations we found our results to be mostly independent of the initialized wave spectrum. Experiments in which the initial random noise perturbation only projected weakly on wavenumbers 2 and 4 showed the same qualitative behaviour as experiments with strong wavenumber 2 and 4 contributions in their initial noise. This further suggests the importance of the scale-selective non-linear interaction which accelerates the growth of waves 2 and 4 via energy transfer from the dominant wave 6.

*To be constructive, would it be possible to show a few additional experiments (initiated with different noise) in Figure 2. (And possibly Figure 4, which shows the final jet states for the same integrations.) I hope that additional solid lines for the \eta=10^-3 experiments would not overly crowd the figure. If all the low noise experiments look exactly the same, the authors could just state this in the text and alleviate my concern from the start.*

In the supplement, we now provide a figure on the evolution of the ensemble members illustrating the small intra-ensemble variability compared to the difference of the evolutions. We further added a reference to this figure at the end of Sec. 2, noting the weak sensitivity to the initial noise configuration.

*Another option would be an additional figure showing that the evolution of key quantities (momentum fluxes, EKE, etc.) follow very similar trajectories for different initializations of noise for all levels of \eta. (Perhaps the variation in noise matter more when \eta is large?) The key is to establish that the difference between lifecycles with different noise realizations is small compared to the difference between the experiments with noise and the monochromatic experiments.*

We agree, this key question is addressed in detail with a figure in the supplement as stated above.

*Minor comments by wavenumber*

*12-3. I found this line to be a bit awkward. Consider "… for LC2 initialisations are found to become unstable eventually, with the onset of instability coming sooner for larger noise perturbations."*

Line is adapted. It now says

*"In particular, the persistent cut-off cyclones that typically form for LC2 initialisations are found to eventually become unstable, with the onset of instability coming sooner for larger noise perturbations.".*

*28 "flavours, or paradigms, of"*

Done.

*Paragraph at 66: As the noise is the major contribution of the manuscript, it might be nice to explain the gist of it in the text. For instance, you could say that the perturbations are white in space, equally exciting all wavenumbers (on average). Perhaps this could be done at line 77 where the amplitude of the noise is introduced.*

Done. The following sentence is inserted in line 77:

*"The noise is white, projecting on average equally on all zonal wave numbers."*

*74 along the same lines, could you briefly characterize the meaning of parameter \hat{U_s} in the text, referring to the equation number in the appendix.*

Done. It now says

*"A set of sensitivity experiments with initial states characterised by varying values of the meridional shear parameter Ûs (traditionally used to trigger LC2, see Eq. A2) showed our results to be overall robust.".*

*77 Appendix*
Corrected.

*90. It was around here that I started worrying whether the realization of the initial noise mattered to the lifecycle. If it does not, a sentence here could put the reader at ease. This could also be discussed in the figure caption.*
Additional to a supplemental figure, the sentence in lines 82-83 was slightly changed to make this clearer from start. The sentence is now:

*"Experiments with $\eta > 0$ are performed as ensembles of several different noise-realisations, although we find the overall large-scale evolution of the life cycles to be insensitive to the particular details of the realisation (see Supplement for details)."*

*107 This is just a comment about style, but I find that footnotes almost over emphasize the point, as the reader breaks off the text to get to it. Consider just putting this material in the main text.*
Thanks for that suggestion, this footnote now entered the main text.

*138. Could you describe this noise induced wave breaking as a secondary instability? The flow is presumably now stable to wave 6 perturbations, but not others?*
Indeed, the growth of the noise could potentially be due to a secondary instability. However, there are several indicators that the increase in EKE for waves 4 and 2 for a second wave breaking is heavily fostered by wave 6 and only partially can be explained by linear theory. One indication for non-linear interactions between the different zonal wave numbers is the deviation in growth rate from the prediction of linear theory, which is already visible during the first days of the simulation. Furthermore, the EKE of wave 6 drops, when wave 4 and 2 reach substantial amplitudes (see e.g. Fig. 6b, day 22). This seems to point to an energy flux from 6 to 4 and 2. Nonlinear processes appear to alter the growth of the noise significantly in all non-monochromatic simulations.
No changes to the text have been made.

*142-145. This line seemed to come too early in the text. Please shift it back after Figure 5a is introduced and the result has been established.*
Done.

*167-7. Is this really similar to quasi-linear non-normal growth? That process is rather distinct from nonlinear wave interactions. Please provide more evidence to support this statement.*
Thank you for pointing this out. We tried to express that the observed growth cannot be explained by single normal mode growth. We do not intend to focus on the distinction of normal vs. non-normal growth, but rather on the non-linear mechanisms still resulting in accelerated, quasi-linear growth. The passage is adapted accordingly (lines 172-173):

*"This quasi-linear accelerated growth suggests a non-linear interaction of the different wave components and could relate to concepts beyond normal growth (comp. e.g., Farrell and Ioannou, 1996)."*

*174. Is "upscale energy cascade" an appropriate way to describe this? Consider "upscale energy transfer" as the flow does not appear to be fully turbulent.*
We agree, and have changed "cascade" to *"transfer"*.

*213. What is "In general" meant to signify here. Is this is reference to the fact that different realizations of noise can lead to different behavior? Or does it refer to difference that occur with variations in the shear parameter \hat{U}_s or other qualities of the initial jet state.*
We have changed the wording to now read:

*"Our life-cycles with noisy initial perturbation ..."*

*218. Same for "typically".*
This word is dropped.

*222 Could you clarify what is meant by "overall net-poleward jet shift periods." The tendency of anticyclonic breaking to shift the jet poleward should be reflected in the mean state.*
*"periods"* is removed. It now says *"overall net-poleward jet shifts"*

*232-4 The meaning of this sentence was a bit obscure to me. Do the authors mean that the remarkably sensitivity of monochromatic experiments to \hat{U}_s, which justified the LC1 vs LC2 paradigms, may not be justified with noise? That is, there aren't really two kinds of wave cycles, but rather a continuum?*
Indeed, describing whole simulations of a longer period including several wave breaking events in different directions with the LC1-LC2 dichotomy might not be justified. For single wave breaking events or phases however, we still deem the paradigms to be useful. In some experiments with only noise as initial perturbation, we even found spatial differences in wave breaking direction at certain times (LC1-like behaviour at some longitudes and LC2-like behaviour at others). In the text we try to indicate that we do not exactly know how to interpret the remaining surf zones indicating the flank of initial wave breaking. We tried to make these aspects clearer now changing the last sentence to (lines 244-246):

*"Growth and decay rates seem to be results of quantitative changes related to the structure of the respective basic state (and hence, e.g., changes in wave propagation properties)."*

*236 and 263. Again, consider "upscale energy transfer"*
Done.

---

## Author Response (AR2)

We thank the referee for reading the revised version of our manuscript and for their constructive comments. In the following we respond to the various comments of each referee and point out any changes we made to the paper based on them. Line numbers and figure references in the reviewer's comments refer to the original manuscript, line and figure references in the responses refer to the revised version. The reviewer's comments are in black; our responses are in blue *blue italics*.

Very minor comments

(1) I think the authors should consider revising the language about wave breaking causing "overall net-poleward jet shifts" (line 229) or that it "dominantly induces poleward jet shifts." (line 279; the final words of the manuscript. Given that there is a stable jet climatology, I don't think it makes sense to say that the eddies are always moving it poleward absent discussion of an opposing process that drives it equator ward.

I appreciate that the authors mean that eddies chiefly transport momentum poleward (out of the subtropics), leading to an observed climate where the jets are poleward of where they would be relative to some radiative equilibrium state absent eddy dynamics. I simply encourage them to emphasize that the eddies are driving the jet poleward against radiative forcing that tends to accelerate them on the equatorward flank; cyclonic wave breaking and a net equatorward momentum transport is a fundamentally transient feature!

*We are now specifically referring to "eddy-induced" jet shifts in Sections 5 and 6.*

(2) Upon reading the paper again, particularly the introduction and the authors' discussion in lines 37-46, it seems that the most robust feature of eddy lifecycles is that higher wave number instabilities tend to favor more cyclonic breaking (as what occurs initially in the high noise LC1 experiments, or standard LC2 experiments with wave 6), and that larger wave numbers tend to break cyclonically (as wave 6 in the monochromatic LC1 experiment, or waves 4, etc. in the noisy LC2 experiments). Adding shear tends to shift the line between anticyclonic and cyclonic wave breaking to higher wave numbers, but in the presence of noise, the anticyclonic features always win in the end. The authors need not make any changes in response to this, but could consider coming back to the point in the discussion or conclusions. Given the large scale gradient in angular momentum from the tropics to extratropics, perhaps larger waves simply have to move momentum poleward?

*We thank the referee for this comment. This is indeed a very important aspect of our findings. We added a paragraph in Section 5 discussing the wave number sensitivity in more detail:*

*In particular, our experiments suggest that the type of canonical LC1/LC2 evolution is mostly set by the dominant zonal wave number of the perturbation. Small wave numbers produce LC1 behaviour, while large wave numbers produce LC2 behaviour (consistent with the findings of Hartmann and Zuercher, 1998}. While for monochromatic initial perturbations small changes in the initial conditions (e.g., adding a meridional surface shear) can induce transitions between LC1 and LC2 evolutions, non-monochromatic initialisations (like used in this study) will always include LC1 phases due to a robust upscale energy transfer (see Fig 7).*

Very minor typographical suggestions

Line 51 "Some main…" seems awkward. Consider "Main findings …"

*Fixed.*

Line 59 "1000 hPa" (no space)

*Fixed.*

Line 68 Consider incorporating this footnote into the text to minimize the disruption, say "are visually similar to the ones used by Thorncraft et al. 1993; who did not specify how their initial conditions were constructed) and hence…"

*We removed the footnote entirely, since it did not actually add any useful information.*